# Thyroid Stimulating Hormone Levels Are Related to Fatty Liver Indices Independently of Free Thyroxine: A Cross-Sectional Study

**DOI:** 10.3390/jcm14072401

**Published:** 2025-03-31

**Authors:** Federica Sileo, Alessandro Leone, Ramona De Amicis, Andrea Foppiani, Laila Vignati, Francesca Menichetti, Giorgia Pozzi, Simona Bertoli, Alberto Battezzati

**Affiliations:** 1International Center for the Assessment of Nutritional Status and the Development of Dietary Intervention Strategies (ICANS-DIS), Department of Food, Environmental and Nutritional Sciences (DeFENS), University of Milan, 20122 Milan, Italy; 2IRCCS Istituto Auxologico Italiano, Clinical Nutrition Unit, Department of Endocrine and Metabolic Diseases, 20145 Milan, Italy; 3IRCCS Istituto Auxologico Italiano, Obesity Unit and Laboratory of Nutrition and Obesity Research, Department of Endocrine and Metabolic Diseases, 20145 Milan, Italy

**Keywords:** thyroid, TSH, fatty liver, steatosis, insulin resistance

## Abstract

**Introduction**: The relationship between metabolic dysfunction-associated steatotic liver disease (MASLD) and thyroid hormones has been established, but the direct effects of TSH on the liver, potentially leading to steatosis, and insulin resistance remain unclear. **Objective**: To investigate the association of TSH levels with MASLD and insulin resistance. **Methods**: We conducted a cross-sectional study of 8825 euthyroid individuals. Subjects were volunteers or patients referred to the International Center for Nutritional Status Assessment (University of Milan, Italy) undergoing clinical examination and blood drawing for thyroid function tests and liver indices calculation. Liver outcomes were fatty liver index (FLI), hepatic steatosis index (HSI), and FIB-4. All associations were adjusted for fT4 and confounders. **Results**: We found a positive association of TSH levels with FLI (β = 2.76; *p* < 0.001) and HSI (β = 0.58, *p* < 0.001). This relationship remained significant when stratifying by sex and BMI category, except for HSI in normal weight individuals. No significant association was found between TSH and hepatic fibrosis or insulin resistance. **Conclusions**: We found a positive association between TSH levels and MASLD in euthyroid individuals independently of fT4, sex, and BMI. Insulin resistance and hepatic fibrosis appear unrelated to TSH, independent of fT4 and BMI. The specific role of TSH in MASLD warrants further investigation.

## 1. Introduction

While thyroid hormones are undoubtedly acknowledged as having a pivotal role in liver metabolism [1], the direct role and function of TSH on the liver has been investigated but is not yet fully understood [2]. The TSH receptor (TSHR) is not only expressed in the thyroid gland, where TSH binding stimulates thyroid hormone production and secretion, but it is also present in extra-thyroidal cells and appears to have pathophysiological relevance. TSHR’s presence in the liver plays a role in lipid metabolism [2,3,4]. While in murine models, TSHR selective deletion in the liver inhibits hepatic lipid accumulation [5], its function in humans has not been completely clarified, as it is not easy to discern TSH signaling effects from those of thyroid hormones. The thyroid hormones triiodothyronine (T3) and thyroxine (T4) stimulate both hepatic lipogenesis and lipolysis by binding to thyroid hormone receptors α (THRα) and β (THRβ), although THRβ is the largely prevalent isoform in the liver [6]. Interestingly, the liver also has receptors for TSH, and some evidence reports that stimulation of these receptors may produce hepatic steatosis [7,8]. Previous studies established an association between TSH levels and MASLD [9,10,11]; this association appears to persist even when considering subclinical hypothyroidism or TSH levels in the upper normal range [12], suggesting the role of TSH in MASLD independently of thyroid hormone levels. In this study, we further investigated the relationship of TSH with MASLD and insulin resistance. Metabolic dysfunction-associated steatotic liver disease (MASLD) [13] is defined as an excess of hepatic lipids (triglyceride content > 5% of organ weight) in subjects with low alcohol consumption (<20 g per day in women and <30 g per day in men) [14]. MASLD represents the hepatic component of metabolic syndrome [15]. Excessive lipid accumulation may lead to inflammation (steatohepatitis) and progressive fibrosis, which can evolve into cirrhosis and hepatocarcinoma [14]. Having more insight into this disease has become of utmost importance given its large diffusion worldwide [16] and scarcity of tools to tackle it.

The gold standard for diagnosis of MASLD is hepatic biopsy. Nonetheless, it is rarely practiced since it is an expensive and invasive procedure [17]. A common tool to assess steatosis is imaging of the liver, especially through ultrasounds; this technique has pitfalls, such as operator dependency and a decreased sensitivity and specificity when liver steatosis is less than 30% [18]. Additionally, imaging techniques may not be available for all individuals of a large sample; in this scenario, non-invasive liver indices become a reliable tool to assess steatosis prevalence, being readily obtainable from biochemical and anthropometric parameters and easy to collect and record in a large database. FLI is a relevant example of a non-invasive fatty liver index with a wide validation on large populations [19]. Additionally, we decided to include HSI, recently validated in reliably predicting this condition [20].

Recently, the relationship between MASLD and non-diabetic endocrinopathies has been investigated, but not enough to outline clear recommendations for the screening and management of endocrinopathies in MASLD [21]. The European Association for the Study of the Liver (EASL), European Association for the Study of Diabetes (EASD), and European Association for the Study of Obesity (EASO) uniquely recommend thyroid function tests and polycystic ovarian syndrome screening within the framework of a more comprehensive work-up of MASLD [19].

Additionally, the relationship between the thyroid and insulin resistance has not yet been completely clarified. Insulin resistance (IR) is defined as the inability of insulin to implement glucose uptake and utilization by peripheral organs, mainly muscles, the liver, and adipose tissue [22,23], resulting in hyperglycemia and hyperinsulinemia [24]. The gold standard for its assessment is hyperinsulinemic euglycemic clamp (HEC), but its complexity has relegated this procedure to a small-scale research tool. A much simpler method to assess insulin resistance for large-scale application is the homeostasis model assessment of insulin resistance (HOMA-IR) [25] using glucose and insulin dosages in fasting conditions with a good correlation coefficient with HEC values. Thyroid function plays a key role in metabolism and body composition, thus potentially affecting IR; according to Ren et al. [26], IR is positively associated with fT3 levels, but they found no correlation with fT4 or TSH, whereas other studies reported an inverse association between IR and TSH levels in euthyroid individuals with normal thyroid ultrasound results [27,28].

In the past, our study group previously examined thyroid function’s influence on metabolism in euthyroid subjects; we found no significant association between thyroid function tests and resting energy expenditure, but we detected a statistically significant positive association between TSH concentrations and triglycerides levels [29].

Given the lack of certainties on this topic due to the few available studies, the highly heterogeneous methodologies, and the contrasting results, we aimed to investigate the association between TSH and MASLD and between TSH and IR in a large sample.

## 2. Materials and Methods

### 2.1. Study Design

From an overall sample of 21,184 subjects, we selected only individuals with complete data of our interest and fulfilling our eligibility criteria, performing a cross-sectional study of 8825 individuals referring to the International Center for Nutritional Status Assessment (ICANS, University of Milan, Milan, Italy) from March 2008 to September 2022. We included healthy volunteers or patients affected with overweight or obesity seeking a nutritional assessment and a personalized diet plan to lose weight. The study was conducted according to the Declaration of Helsinki guidelines of 1964. Written informed consent was obtained from all participants. Inclusion criteria were ≥18 years of age, BMI ≥ 18.5 kg/m^2^, fT4 levels within laboratory reference ranges (7–20 pg/mL), and TSH levels compatible with euthyroidism (0.5–4.5 µUI/mL) according to previously established thresholds [30]. Exclusion criteria were medical history of alcoholic hepatitis, hepatitis C virus infection, hepatitis B virus infection, liver cirrhosis, known history of hyperthyroidism or hypothyroidism, levothyroxine treatment, and antithyroid drugs use. Unfortunately, we could not exclude a previous infection by hepatitis viruses by an assessment of antibody status given the absence of serology in routine lab tests performed at our institutional center. Our exclusion criteria directly pertained to liver dysfunction or conditions capable of interfering with thyroid function; therefore, the selected population could potentially include individuals with diseases of other organs or systems.

### 2.2. Physical Examination and Anthropometric Measurements

All subjects underwent a clinical examination, where past and present medical history were investigated, and a physical examination was performed. The same operator took anthropometric measurements according to international guidelines [31]. Measurement of weight was performed with an electronic scale with an accuracy of 100 gr (Seca 700, Seca Corporation, Hamburg, Germany), height was measured with a vertical stadiometer with an accuracy of 0.1 cm, and waist circumference (WC) was measured using a nonelastic tape at the midpoint between the iliac crest and the last rib with an accuracy of 0.5 cm. A skinfold caliper (Holtan Ltd., Crymych, Wales) was used to measure the four skinfolds (biceps, triceps, subscapular, and suprailiac). Fat mass was calculated using the Durnin–Womersley equation [32], and percent fat mass (body fat %) was calculated as (fat mass (kg)/body weight (kg)) × 100.

### 2.3. Laboratory Measurements

Blood samples were drawn in fasting conditions between 08:30 and 09:00 a.m. for the measurement of blood glucose, insulin, triglycerides, alanine transaminase (ALT), aspartate transaminase (AST), and gamma-glutamyl-transferase (GGT), essential for HOMA-IR index and steatotic liver indices calculation, TSH, and fT4. In a subset of 2152 patients, platelet measurements were available, allowing us to calculate FIB-4 index. All samples were drawn at the same time of day in order to minimize circadian oscillations of TSH and fT4 particularly. Plain tubes were centrifuged, and serum was stored at −80 °C.

Insulin, TSH, and fT4 were measured using the immunoenzymatic method (Cobas E 411, Roche Diagnostics, Rotkreuz, Switzerland). Triglycerides, glucose, ALT, AST, and GGT were measured using an enzymatic method (Cobas Integra 400 Plus, Roche Diagnostics, Rotkreuz, Switzerland). Platelets were measured using the hemochromocytometric method (Sysmex XN-550, Norderstedt-Hamburg, Germany).

### 2.4. Liver Steatosis and Fibrosis and Insulin Resistance Assessment

Non-invasive liver indices, used to evaluate hepatic steatosis and fibrosis probability, were the following:Fatty liver index (FLI) [33]: eLP/(1 + eLP) × 100 where LP (linear predictor) = 0.953 × ln(triglycerides (mg/dL)) + 0.139 × BMI (kg/m^2^) + 0.718 × ln (GGT (U/L)) + 0.053 × WC (cm) − 15.745;Hepatic steatosis index (HSI) [34] = 8 × (ALT (U/L)/AST (U/L)) + BMI (kg/m^2^) + 2 if woman + 2 if diabetes mellitus;FIB-4 index [35]: Age ([years] × AST [U/L])/((PLT [10(9)/L]) × √(ALT [U/L])).

Insulin resistance was evaluated using the HOMA-IR index [36], which is calculated by using the following formula: fasting glucose (mg/dL) × fasting insulin (mU/L)/405.

An FLI ≥ 60 and an HSI > 36 are highly indicative of fatty liver disease, as established in their validation studies; an FIB-4 > 2.67 is highly indicative of fibrosis; HOMA-IR > 2.5 is indicative of insulin resistance [33,34,36,37].

### 2.5. Statistical Analysis

Given the non-normal distribution of continuous variables, the descriptive statistics results are presented as the median and interquartile range. Categorical variables are presented as frequencies and percentages. A linear regression model adjusted for non-normal distribution was used to assess the association between TSH and non-invasive liver indices (FLI, HSI, and FIB-4) and TSH and insulin resistance (HOMA-IR). We chose covariates for our models based on biological confounding plausibility. Potential confounders included sex, age, fT4, and smoking habit at blood sampling for FLI and his as well as sex, age, BMI, fT4, and smoking habit at blood sampling for FIB-4 and HOMA-IR. The two-sided significance level was set at *p* < 0.05. All statistical analyses were performed using StataMP 18.0 64-bit (Stata Corporation, College Station, TX, USA; RRID:SCR_012763).

## 3. Results

Epidemiologic, anthropometric, and biochemical characteristics of our population are represented in Table 1.

The association of TSH levels with steatotic liver indices in the whole population is reported in Table 2.

In our subjects, we found a positive association between TSH levels and steatotic liver indices, with a different magnitude of association comparing the two indices. A 1-point increase in TSH was associated with a 2.76 increase in FLI and a 0.58 increase in HSI.

In view of the impact of the patient’s weight on both these indices and TSH levels, we stratified this analysis by BMI category, as displayed in Table 3. We found that this association remained statistically significant for people with a BMI compatible with overweight and obesity, i.e., >25 kg/m^2^ (2.17 FLI increase, *p* < 0.001, for 1-unit increase in TSH; 0.45 HSI increase, *p* < 0.001, for 1-unit increase in TSH). The association was also statistically significant for normal weight individuals (BMI 18.5–24.9 kg/m^2^) concerning FLI (0.61 FLI increase, *p* = 0.02, for 1-unit increase in TSH) but not HSI.

In addition, considering the sexual dimorphism of liver disease and thyroid dysfunction, we repeated the same analysis after stratifying by sex. This positive association remained true for both men and women, even if with different magnitudes between the two sexes (3.22 FLI increase, *p* < 0.001, for 1-unit increase in TSH; 0.62 HSI increase, *p* < 0.001, for 1-unit increase in TSH in women; 1.51 FLI increase, *p* = 0.02, for 1-unit increase in TSH; 0.45 HSI increase, *p* = 0.01, for 1-unit increase in TSH in men), as represented in Appendix A.

We performed the same analyses with the FIB-4 index in the subgroup of our population with platelet measurements, but we found no statistically significant correlation of this index with TSH levels, even when stratifying by sex (see Appendix A). It should be noted that FIB-4 index was calculated in a subset of 2152 out of 8825 patients. This subset was comparable to the comprehensive cohort of 8825 patients in terms of anthropometric characteristics and sexes distribution.

Finally, we investigated the association between TSH levels and insulin resistance, which was not significant in our population, as shown in Table 4. This result was confirmed when performing the same analysis after stratifying by sex (Appendix A) and by BMI category (Appendix A).

## 4. Discussion

The thyroid gland has a profound effect on liver function, lipid metabolism, and insulin sensitivity. Therefore, our study tried to explore these complex interactions by investigating thyroid function tests and fatty liver probability in a very large population while adjusting for potential confounding factors. We investigated the association of TSH with MASLD probability expressed through three of the main non-invasive fatty liver indices used in clinical practice, FLI, HSI, and FIB-4, and the potential role of TSH in the modulation of insulin sensitivity, evaluated through HOMA-IR. In our analysis, we included people having normal fT4 levels and TSH levels compatible with euthyroidism. We found a positive association between TSH levels and both FLI and HSI; when stratifying our population by sex and BMI category, this relationship was confirmed, except for HSI in the normal weight category.

Our interest in TSH signaling and MASLD comes from the discovery of TSHR expression in multiple organs other than the thyroid gland, which led to the hypothesis that TSH function not only stimulates the thyroid but has pleiotropic functions and may play a role in several human diseases.

The first hypothesis of a potential role of TSH–TSHR signaling in the liver comes from the discovery of the TSHR mRNA transcript in hepatocytes [38]. Nonetheless, the first interpretation of this finding was that this could be an illegitimate transcription, i.e., low transcription of a tissue-specific gene in nonspecific cells. Then, Zhang et al. [39] demonstrated that TSHR is present in hepatocytes’ cell membrane and is also functional. Furthermore, a liver-specific deletion of TSHR in a murine model demonstrated that hepatic TSHR modulation is deeply involved in lipid metabolism [5]. Ultimately, the potential role of TSH signaling in MASLD has been hypothesized because increased mitochondrial oxidative stress, involved in MASLD pathogenesis, has been observed in murine models to be increased by TSH-hepatic TSHR signaling [40].

These in vivo studies paved the way for exploring this relationship also in humans. However, investigating the pathophysiological role of TSH in epidemiological studies is difficult for several reasons: distinguishing the consequences of TSH elevation and thyroid hormones’ deficiency and vice versa, given their reciprocal relationship, is very challenging; TSHR may be expressed at very low levels in non-thyroidal tissues; and we cannot exclude a minimal amount of peripheral TSH production [41].

Epidemiological studies in human cohorts have observed a link between MASLD and hypothyroidism, where the main role of thyroid hormones in lipid metabolism (lipid export and oxidation) and hepatic insulin sensitivity has been proven on multiple occasions [1,42].

Four recent systematic reviews and meta-analyses have found contrasting results regarding the association between MASLD and thyroid function [43,44,45,46]. These divergent results may be due to different reasons: the use of different eligibility criteria, selection biases, definition of outcomes, different quality assessments, and heterogeneous methodologies. Nonetheless, an eventual causality could not be established due to the observational nature of the studies. In 2021, a Chinese study performed a genome-wide association study with Mendelian randomization to assess the putative causal effect of hypothyroidism on MASLD, and they were able to demonstrate such causality [47]. Furthermore, an interesting longitudinal study published in 2021 reported that having higher TSH levels led to a higher incidence rate of MASLD over a median follow-up of 4.03 years [11]. Regrettably, in this case, we have no data on contextual fT4 levels that would allow us to distinguish between thyroid hormone and TSH’s effects on the liver. Our study provides data that support the hypothesis that TSH has a direct role in modulating intrahepatic lipid metabolism, even in the range of euthyroidism and independently of fT4 levels.

In our study, we also investigated insulin resistance and TSH levels, finding no significant association between TSH levels and insulin resistance calculated as HOMA-IR, both when considering the comprehensive population and when performing the same analysis separately on men and women and on different BMI categories, normal weight, and overweight/obesity.

We evaluated this correlation because TSH-hepatic TSHR signaling is also involved in glucose metabolism; TSH increases gluconeogenesis by enhancing hepatic cAMP-regulated transcriptional coactivator-2 expression, which in turn stimulates glucose-6-phosphatase and cytosolic phosphoenolpyruvate carboxykinase expression. These two compounds promote decarboxylation of oxaloacetate to phosphoenolpyruvate and hydrolyzation of glucose-6-phosphate into glucose and inorganic phosphate [48]. Hence, we may infer that higher TSH levels lead to insulin resistance, type 2 diabetes, and an increased risk of MASLD, and some authors found a positive linear association between TSH in euthyroid range and insulin resistance in both diabetic and non-diabetic patients [49]. Unfortunately, we could not confirm this evidence in our cohort. Similarly, another study by Muscogiuri et al. [50] found in its cohort comparable results, concluding that the best predictor of TSH levels in patients with obesity was visceral adipose tissue rather than insulin resistance, measuring these two entities with a CT scan and euglycemic clamp, respectively. Indeed, TSH levels were found to be associated with other factors concurring to a metabolic syndrome diagnosis, but not specifically with insulin resistance evaluated as HOMA-IR, as investigated in a cohort of euthyroid postmenopausal women [51]. In two other cross-sectional studies, they documented a significant association of low normal fT4 levels with insulin resistance measured as HOMA-IR but did not find the same statistically significant association with TSH levels [52,53]. Ultimately, a prospective observational study by Ferrannini et al. corroborated the hypothesis that thyroid hormone levels are tightly correlated with insulin resistance measured by euglycemic clamp, underlining the positive relationship of fT3 with insulin resistance but not TSH levels [54].

Our findings could have relevant clinical implications. The link between MASLD and thyroid function has plenty of pathophysiological evidence, thus contributing to the recent FDA approval of the drug resmetirom [55], a selective THRβ agonist, for MASLD. The drug’s selectivity for the target receptor is crucial to mitigate the levothyroxine-induced issue of binding to both THRα and THRβ, which would otherwise lead to hyperthyroidism in non-target tissues. Nonetheless, data from several studies together with our results from a large cohort led us to speculate that high-normal TSH levels could affect liver metabolism despite normal fT4 levels, with important clinical implications for MASLD prevention and management. This suggests that TSH assessment should be included in the standard diagnostic-therapeutic approach to MASLD; this approach should consist of a comorbidities evaluation, tailored nutritional counseling, and a profound lifestyle change.

As already mentioned, the difficulties of evaluating TSH’s effects in hepatic lipid metabolism lay in the complexity of distinguishing thyroid hormones and TSH’s effects; if this is challenging in murine models, this is even more challenging in epidemiological studies on humans. We performed a multivariate linear regression considering different potential confounding factors, starting with fT4 levels. Indeed, we chose to exclude people with fT4 levels outside the normal range, and we included fT4 as a covariate, confirming a statistically significant association between TSH levels and non-invasive fatty liver indices. Other covariates included in the analysis were sex, age, and smoking status. We deliberately chose not to include BMI, triglycerides, waist circumference, liver enzymes, or other factors strictly linked to metabolic syndrome because they were among the factors used for the calculation of FLI or HSI. We instead included BMI as a covariate in linear regression estimation for FIB-4 and HOMA-IR, since it is not included in these indices’ calculations and strongly impacts both of these indices. Considering the most appropriate covariates is of utmost importance when assessing an association between two factors; hence, we accurately chose them for our multivariate linear regression analysis.

Our work has the following strengths: firstly, the data came from a large population. Moreover, we used externally validated non-invasive liver indices to predict MASLD probability with similar predictive abilities compared to the more frequently used ultrasound imaging, according to the most recent guidelines for epidemiological studies [19]. Third, we accurately controlled our results for specific covariates, particularly fT4 levels, which allowed us to examine the independent role of TSH-TSHR signaling in hepatic lipid metabolism.

However, our study also has some limitations. First, the cross-sectional design of the study prevented us from establishing a causal relationship, for which prospective studies are needed to thoroughly investigate this correlation. Second, the lack of serum fT3 concentrations, not routinely assessed in our study population, may represent a limit of our study. However, fT4 is the main substrate for peripheral conversion into fT3 and the predominant form of thyroid hormone secretion. Moreover, serum fT3 concentrations are only a part of the fT3 amount, not providing insight into the actual amount of bioactive fT3 acting at the peripheral tissue level, which is generated intracellularly and not directly measurable. In light of these reflections, we deemed fT4 a reliable and comprehensive indicator of thyroid hormone status. Third, the use of FLI and HSI, even if they are externally validated indices, does not have a 100% correspondence with the MASLD gold standard diagnosis, which is biopsy; the same is true for FIB-4 in liver fibrosis assessment. However, liver biopsy is performed very rarely for the above examined reasons, and almost no studies use it for assessing MASLD. Additionally, non-invasive liver indices represent a valid option for liver health screening in large epidemiological studies in the absence of imaging reports [19]. Finally, our sample was only composed of Caucasian individuals, thus considerably limiting the generalizability of our results to Western countries and leaving the need for further investigations in individuals of other ethnicities.

## 5. Conclusions

In conclusion, we found a positive association between TSH levels and MASLD probability independently of fT4 levels, regardless of sex and BMI, except for HSI in normal weight individuals. TSH levels might not influence hepatic fibrosis probability and insulin resistance, independently of fT4 levels and BMI. All these findings would need further investigations through longitudinal studies to establish a cause–effect relationship and thus eventually define that not only thyroid hormones but also TSH levels are worthy of evaluation during MASLD management since they may contribute to its onset or exacerbation.

## Figures and Tables

**Table 1 jcm-14-02401-t001:** Characteristics of patients.

	Total N = 8825		
	N	%	
Sex (male)	2648	33.1	
BMI category			
Normal weight	2026	23	
Overweight	3358	38	
Class I obesity	2135	24	
Class II obesity	867	10	
Class III obesity	439	5	
Smoking status			
Non-smoker	4372	49.5	
Ex-smoker	1746	19.8	
Smoker	2707	30.7	
	**Median**	**P25**	**P75**
Age (years)	46	37	54
BMI (kg/m^2^)	28.45	25.31	32.44
Body fat (%)	37.2	32.4	40.8
Waist circumference (cm)	96.1	86.5	106.7
TSH (µUI/mL)	1.88	1.36	2.54
fT4 (pg/mL)	11.7	10.6	12.9
Triglycerides (mg/dL)	90	65	130
GGT (U/L)	18.5	12.7	29
Serum glucose (mg/dL)	95	89	103
Insulin (U/L)	9.86	6.81	14.94
HOMA-IR	2.33	1.58	3.73
ALT (U/L)	19	14	28
AST (U/L)	18.7	15.6	22.9
FLI	45.4	17.2	78.9
HSI	40.1	35.9	45.3

Abbreviations: TSH, thyrotropin stimulating hormone; fT4, free thyroxine; BMI, body mass index; GGT, gamma glutamyl-transferase; HOMA-IR, Homeostatic Model Assessment for Insulin Resistance; ALT, alanine transaminase; AST, aspartate transaminase; FLI, fatty liver index; HSI, hepatic steatosis index.

**Table 2 jcm-14-02401-t002:** Association of TSH levels with steatotic liver indices in our population.

	Coefficient (β)	Standard Error	T Value	*p* Value
FLI	2.76	0.36	7.67	<0.001
HSI	0.58	0.09	6.2	<0.001

β values are linear regression coefficients associated with a one-point increase in TSH. Models are adjusted for age, sex, fT4 levels, and smoking status. Abbreviations: FLI, fatty liver index; HSI, hepatic steatosis index.

**Table 3 jcm-14-02401-t003:** Association of TSH levels with steatotic liver indices in our population stratified by BMI category.

	Normal Weight				Overweight and Obesity			
	Coefficient (β)	Standard Error	T Value	*p* Value	Coefficient (β)	Standard Error	T Value	*p* Value
FLI	0.61	0.26	2.35	0.02	2.17	0.37	5.92	<0.001
HSI	0.07	0.08	0.91	0.36	0.45	0.09	5.04	<0.001

β values are linear regression coefficients associated with a one-point increase in TSH. Models are adjusted for age, sex, fT4 levels, and smoking status. Normal weight includes individuals with BMI between 18.5 and 24.9 kg/m^2^; overweight and obesity includes individuals with BMI ≥ 25 kg/m^2^. Abbreviations: FLI, fatty liver index; HSI, hepatic steatosis index.

**Table 4 jcm-14-02401-t004:** Association of TSH levels with insulin resistance in our population.

	Coefficient (β)	Standard Error	T Value	*p* Value
HOMA-IR	−0.04	0.04	−1.12	0.26

β value is linear regression coefficient associated with a one-point increase in TSH. Models are adjusted for fT4 levels, age, sex, BMI, and smoking status. Abbreviations: HOMA-IR, Homeostatic Model Assessment for Insulin Resistance.

## Data Availability

Due to patient privacy regulations, the raw data underlying this study cannot be publicly shared. Anonymized data may be made available upon reasonable request to the corresponding author, subject to ethical review and approval.

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
