# Peer review of "Thyroid Stimulating Hormone Levels Are Related to Fatty Liver Indices Independently of Free Thyroxine: A Cross-Sectional Study"

_jcm, 2025, doi:10.3390/jcm14072401_

Round 1

Reviewer 1 Report

Comments and Suggestions for Authors

In this study, the authors’ aim was to evaluate the associations between thyroid hormone levels, body composition, MASLD and insulin resistance in healthy euthyroid subjects. I have several comments and suggestions which I would like to point out. 

Introduction

  1. The introduction is very long, I suggest shortening it. 
  2. The content of the text in lines 89-94 is a repetition of what is already presented in lines 47-50. To enhance readability and ensure the message remains clear, the text in lines 89-94 should be positioned directly after lines 47-50. This will make the writing more cohesive and avoid unnecessary repetitions. 
  3. “IR was found positively associated with fT3 levels in a previous study by Ren et al. [30] and also inversely associated with TSH levels in euthyroid individuals with normal thyroid ultrasound results [31,32]. Nonetheless, Ren et al. found no correlation between IR and fT4 or TSH [30].”- the text requires clarification, for example: According to Ren et al. IR was found positively associated with fT3 and not with TSH [30] whereas other studies have reported an inverse association between IR and TSH levels [31, 32].
  4. After accurately reading the introduction, I concluded that the relationship between TSH and MASLD already exists and is well-established. However, it is unclear what new insights the researchers were aiming to add to the existing knowledge. The aim of the study is not clearly articulated.

Methods

1.The authors did not clearly explain the inclusion and exclusion criteria. Based on the provided criteria, only euthyroid patients without liver disease were included in the study. Were individuals with hypertension, hyperlipidemia, or a history of cardiovascular disease included in the study? What about other diseases?

Results

  1. Table 1 should be improved. Anthropometric and demographic data should be presented before the laboratory data for better clarity and logical flow. 

2.Do the authors have data on the distribution of patients across three classes of obesity? If available, this information should be included in table 1. 

  1. I suggest considering better ways to present tables 2 and 3 more clearly.

Discussion

  1. In line 264, it is unclear what references 40 and 41 refer to.
  2. In line 278, the authors mention five systematic reviews and meta-analyses, but only four references are listed at the end of the sentence.
  3. I suggest that the authors include in the references-A study by Zhu et al. which showed a positive and linear association between TSH levels within reference range and HOMA-IR in both non-DM subjects and type 2 DM patients (Zhu P, Liu X, Mao X. Thyroid-Stimulating Hormone Levels Are Positively Associated with Insulin Resistance. Med Sci Monit. 2018 Jan 17;24:342-347).  
  4. ”Third, we accurately controlled our results for specific covariates, in particular we controlled for fT4 levels, that allowed to examine the independent role of TSH-TSHR signaling in hepatic lipid metabolism”- I cannot agree with the authors’ conclusion. As far as we know, FT4 is converted into FT3 (the bioactive hormone) in the periphery under the influence of deiodinases. However, FT3 was not measured in this study, which does not allow examining the independent role of TSH-TSHR.
  5. There are several important comments and potential limitations in this study: FT3 was not measured, and the study participants were divided into two groups: normal weight and overweight/obese. By classifying the patients into obesity classes they could have provided more accurate and clinically relevant results 
  6. For their study the authors selected healthy participants.  It is unclear what percentage of participants there were, according to the FLI and HSI, that were diagnosed with MASLD and FIB-4 with fibrosis? This information seems important and relevant for the study as well.

Author Response

For research article

Response to Reviewer 1 Comments

1. Summary

Thank you for taking the time to review this manuscript. Please find the detailed responses below and the corresponding revisions/corrections in track changes in the re-submitted files.

2. Questions for General Evaluation

Reviewer’s Evaluation

Response and Revisions

Does the introduction provide sufficient background and include all relevant references?

Must be improved

Dear reviewer, thank you for your valuable feedback. Please, find our point-by-point response below.

Is the research design appropriate?

Can be improved

Are the methods adequately described?

Can be improved

Are the results clearly presented?

Must be improved

Are the conclusions supported by the results?

Can be improved

3. Point-by-point response to Comments and Suggestions for Authors

Comments 1:

In this study, the authors’ aim was to evaluate the associations between thyroid hormone levels, body composition, MASLD and insulin resistance in healthy euthyroid subjects. I have several comments and suggestions which I would like to point out.

Dear Reviewer 1,

thank you very much for your comments concerning our manuscript entitled " TSH levels are related to fatty liver indices independently of FT4: a cross-sectional study." (Submission ID: jcm-3540666). We studied comments carefully and have made corrections which we hope meet with approval. Revised passages are in the tracked changes version of our manuscript. The response to the comments and the main corrections in the paper are given as follows:

Introduction

1.      The introduction is very long, I suggest shortening it.

Thank you very much for your comment, we shortened this section as much as possible. The cut sections remained as “tracked changes” in the manuscript.

2.      The content of the text in lines 89-94 is a repetition of what is already presented in lines 47-50. To enhance readability and ensure the message remains clear, the text in lines 89-94 should be positioned directly after lines 47-50. This will make the writing more cohesive and avoid unnecessary repetitions.

Thank you for your suggestion, we moved the passage accordingly to make the writing more fluent.

3.      “IR was found positively associated with fT3 levels in a previous study by Ren et al. [30] and also inversely associated with TSH levels in euthyroid individuals with normal thyroid ultrasound results [31,32]. Nonetheless, Ren et al. found no correlation between IR and fT4 or TSH [30].”- the text requires clarification, for example: According to Ren et al. IR was found positively associated with fT3 and not with TSH [30] whereas other studies have reported an inverse association between IR and TSH levels [31, 32].

Thank you for your insightful comment; we rephrased the sentence following your suggestions in order to make the passage clearer, which now reads: “according to Ren et al.[30], IR was positively associated with fT3 levels but they found no correlation with fT4 or TSH, whereas other studies reported an inverse association between IR and TSH levels in euthyroid individuals with normal thyroid ultrasound results[31,32].” (page 3, line 118).

4.      After accurately reading the introduction, I concluded that the relationship between TSH and MASLD already exists and is well-established. However, it is unclear what new insights the researchers were aiming to add to the existing knowledge. The aim of the study is not clearly articulated.

Your feedback has been helpful in revising and improving our manuscript. Indeed, basic science studies established a direct relationship of TSH with MASLD, but this association has been clinically explored only to a limited extent, hence literature is still incomplete and contradictory on the topic, with conflicting results. We have made revisions based on your kind suggestions and comments and have rephrased the aim of the study more appropriately to further emphasize this belief within the manuscript (the passage at page 3, line 124, now reads: “Given the lack of certainties on this topic due to the few available studies, the highly heterogeneous methodologies, and the contrasting results, we aimed to investigate the association between TSH and MASLD and between TSH and IR in a large sample”).

Methods

1.      The authors did not clearly explain the inclusion and exclusion criteria. Based on the provided criteria, only euthyroid patients without liver disease were included in the study. Were individuals with hypertension, hyperlipidemia, or a history of cardiovascular disease included in the study? What about other diseases?

Thank you for pointing this out, we agree with this comment. Our population is quite heterogeneous, both in terms of demographic and anamnestic characteristics. We deemed it appropriate to exclude patients with conditions directly related to the variables of interest in our study, namely hepatic or thyroid disorders. Therefore, the selected population could potentially include individuals with other diseases, as long as these were not directly associated with liver dysfunction or capable of interfering with thyroid function. We added this explanation along the manuscript in order to make the passage clearer (page 4, line 146), which now reads: ”Our exclusion criteria directly pertain liver dysfunction or conditions capable of interfering with thyroid function, therefore the selected population could potentially include individuals with diseases of other organs or systems.”.   

Results

1. Table 1 should be improved. Anthropometric and demographic data should be presented before the laboratory data for better clarity and logical flow.

Thank you for your useful suggestion, we modified Table 1 accordingly.

2. Do the authors have data on the distribution of patients across three classes of obesity? If available, this information should be included in table 1.

Thank you for your suggestion, we have the distribution of patients across the three classes of obesity available, therefore we included such data in Table 1 as you proposed.

2.      I suggest considering better ways to present tables 2 and 3 more clearly.

Thank you for your valuable feedback. We added in Tables 2 and 3 standard error and T value for each regression analysis. We hope these changes address your concerns and provide a clear representation of performed analyses.

Discussion

1.      In line 264, it is unclear what references 40 and 41 refer to.

Thank you very much for pointing this out. It is a typo resulting from a previous version of the article. We have, accordingly, removed these two references.

2.      In line 278, the authors mention five systematic reviews and meta-analyses, but only four references are listed at the end of the sentence.

Thank you for your comment, we changed the number from five to four, as you kindly pointed this out.

3.      I suggest that the authors include in the references-A study by Zhu et al. which showed a positive and linear association between TSH levels within reference range and HOMA-IR in both non-DM subjects and type 2 DM patients (Zhu P, Liu X, Mao X. Thyroid-Stimulating Hormone Levels Are Positively Associated with Insulin Resistance. Med Sci Monit. 2018 Jan 17;24:342-347). 

Thank you for your helpful suggestion, the cited article has been added among our references, as per your recommendation.

4.      ”Third, we accurately controlled our results for specific covariates, in particular we controlled for fT4 levels, that allowed to examine the independent role of TSH-TSHR signaling in hepatic lipid metabolism”- I cannot agree with the authors’ conclusion. As far as we know, FT4 is converted into FT3 (the bioactive hormone) in the periphery under the influence of deiodinases. However, FT3 was not measured in this study, which does not allow examining the independent role of TSH-TSHR.

We agree with your observation. Indeed, fT3 is the biologically active hormone, primarily derived from the peripheral conversion of fT4 via deiodinase enzymes. Unfortunately, fT3 was not routinely assessed in our study population, and therefore we were unable to include it in our analysis. However, fT4 represents the main substrate for peripheral conversion into fT3 and constitutes the predominant form of thyroid hormone secretion. Moreover, serum fT3 concentrations primarily reflect the fraction directly secreted by the thyroid gland and that deriving from liver and kidney conversion, without providing insight into the actual amount of bioactive fT3 acting at the peripheral tissue level, generated intracellularly and not directly measurable. Given these arguments, we considered fT4 a reliable and comprehensive surrogate marker of thyroid hormone status. Therefore, adjusting our results for fT4 levels appears to be a reasonable approach to account for the independent role of TSH-TSHR signaling.

In agree with your comment, we further pointed out this limitation in the discussion section at page 9, line 370 (please see the passage: “Second, the lack of serum fT3… surrogate marker of thyroid hormone status”).

5.      There are several important comments and potential limitations in this study: FT3 was not measured, and the study participants were divided into two groups: normal weight and overweight/obese. By classifying the patients into obesity classes they could have provided more accurate and clinically relevant results

Thank you for pointing this out. As mentioned in the previous point, unfortunately fT3 was not routinely assessed in our population. We highlighted this point along with the manuscript and included the rationale why the sole adjustment of our regression analyses for fT4 levels could be reasonable. We followed your advice of classifying the patients into obesity classes and performed again the analyses, as per your recommendations, additionally performing a sensitivity analysis. When further stratifying into overweight and the three obesity classes, we can still observe the trend of the positive association between TSH levels and liver indices, but we lose statistical significance in specific subcategories because of loss of statistical power. Specifically, we obtained statistically significant positive associations between FLI and I class (β=0.78; p=0.05) and III class obesity (β=0.44; p=0.04) and a marginal significance of HSI in III class obesity (β=0.45; p=0.09).  

6.      For their study the authors selected healthy participants.  It is unclear what percentage of participants there were, according to the FLI and HSI, that were diagnosed with MASLD and FIB-4 with fibrosis? This information seems important and relevant for the study as well.

Thank you for pointing this out. We decided to consider FLI, HSI, and FIB4 as continuous variables in our manuscript, both in the summary table of characteristics of the study population and in the regression analysis, since we consider those diseases as a wide spectrum of alterations and because they indicate a probability of disease rather than a diagnosis, as in imaging techniques or biopsy. However, we applied cutoffs of FLI, HSI, and FIB4 taken from validation studies and found the following percentages of positivity:

- FLI = 3891/8825 (44%)

- HSI = 6547/8825 (74%)

- FIB4 = 14/2078 (0.7%)

As evidenced by the reported percentages, the cut-off values of tools for assessing hepatic steatosis and fibrosis exhibit different sensitivity and specificity. This variability arises from the construction principle of these tools and the parameters included in their respective formulas, which inherently differ. In our study, we addressed these discrepancies by considering these variables as continuous, thereby circumventing the limitations associated with discrete cut-off values.

Reviewer 2 Report

Comments and Suggestions for Authors

The Authors present a study about the possibile association of TSH level and MASLD not depending of T4 level.

I found the work well presented and developed. However, some suggestions and corrections could be made.

1)"a direct role and function of TSH on liver has been just recently 35
investigated and not yet fully understood." I suggest to provide at least a reference justifying the sentence.

2) " In the past, our study group already examined thyroid function influence on metabolism, focusing on resting energy expenditure and different cardiovascular risk factors in euthyroid subjects: we found no significant association between thyroid function tests and resting energy expenditure measured using a canopy-equipped indirect calorimeter, but we detected a statistically significant positive association between TSH concentrations and triglycerides levels[8]. In this study we further investigated the relationship of TSH with MASLD and insulin resistance" I suggest to traspose this part at the end of "Introduction".

3 "The study was conducted according to the Declaration of Helsinki guidelines in 1964." I believe an Etical Approval is also required, please provide this.

4) Lines 125-126: why this TSH value for euthyroidism, when 0.4 mIU/L is usually used as the cut-off for subclinical hyperthyroidism? Reference 33 does not justify this choice either. Please explain.

5) Line 128: what is mean "Thyroid disfunction"? Please specify (I think patients with TSH  out of range, is that right?).

6) Line 278-279:  You cited five studies, but the references were only four. Furthermore, how do you explain that four systematic reviews and meta-analyses on the same document can give different results? I suggest to give your interpretation in the discussion.

7) Given your findings, a hot question: what do you think about the possibility of levotiroxine therapy to reduce TSH levels in the prevention of MASLD? I suggest that you add your opinion in the "Discussion" section, possibly including a suggestion for further studies.

Author Response

For research article

Response to Reviewer 2 Comments

1. Summary

Thank you very much for taking the time to review our manuscript entitled "TSH levels are related to fatty liver indices independently of FT4: a cross-sectional study." (Submission ID: jcm-3540666). We all appreciate you greatly for your

positive comments and recognition of our work. We studied comments carefully and

have made corrections accordingly. Revised portions are in tracked changes in our manuscript. Please find detailed responses below.

2. Questions for General Evaluation

Reviewer’s Evaluation

Response and Revisions

Does the introduction provide sufficient background and include all relevant references?

Can be improved

Dear reviewer, thank you for your valuable feedback. Please, find our point-by-point response below.

Is the research design appropriate?

Yes

Are the methods adequately described?

Can be improved

Are the results clearly presented?

Yes

Are the conclusions supported by the results?

Yes

3. Point-by-point response to Comments and Suggestions for Authors

The Authors present a study about the possible association of TSH level and MASLD not depending of T4 level.

I found the work well presented and developed. However, some suggestions and corrections could be made.

1)"a direct role and function of TSH on liver has been just recently 35

investigated and not yet fully understood." I suggest to provide at least a reference justifying the sentence.

Thank you very much for your suggestion, we added a reference which justifies our sentence.

2) " In the past, our study group already examined thyroid function influence on metabolism, focusing on resting energy expenditure and different cardiovascular risk factors in euthyroid subjects: we found no significant association between thyroid function tests and resting energy expenditure measured using a canopy-equipped indirect calorimeter, but we detected a statistically significant positive association between TSH concentrations and triglycerides levels[8]. In this study we further investigated the relationship of TSH with MASLD and insulin resistance" I suggest to traspose this part at the end of "Introduction".

Thank you for your useful suggestion, we moved this passage at the end of Introduction section, as per your recommendation.

3 "The study was conducted according to the Declaration of Helsinki guidelines in 1964." I believe an Etical Approval is also required, please provide this.

Thank you for your comment, editors also requested the document of approval by the institutional ethical committee, which we sent them via email and included in the submission form. This was the additional sentence about ethical committee approval:

“Full ethic institutional name is the Ethics Committee of the University of Milan (reference number and date of ethical approval: n 23/2016, July 18, 2016). Written informed consent was obtained from all participants.”

4) Lines 125-126: why this TSH value for euthyroidism, when 0.4 mIU/L is usually used as the cut-off for subclinical hyperthyroidism? Reference 33 does not justify this choice either. Please explain.

Thank you for your insightful comment. It was a typo, we corrected the typo in the revised version of the manuscript (page 4, 2.1 section of materials and methods, line 146).

5) Line 128: what is mean "Thyroid disfunction"? Please specify (I think patients with TSH  out of range, is that right?).

Thank you for your comment. Such exclusion criteria refer to medical history of thyroid dysfunction (hyperthyroidism of hypothyroidism) according to our registry, since our database has both biochemical parameters and medical history records of our study population. We specified this in the passage, which now reads: “known history of thyroid hyperthyroidism or hypothyroidism” (page 4, 2.1 section of materials and methods, line 149).

6) Line 278-279:  You cited five studies, but the references were only four. Furthermore, how do you explain that four systematic reviews and meta-analyses on the same document can give different results? I suggest to give your interpretation in the discussion.

Thank you for your comment, we changed the number from five to four, as you kindly pointed this out. We also added our personal interpretation of discrepant results among the different studies, as per your recommendation (page 8, discussion section, line 302: “These divergent results may be due to different reasons: the use of different eligibility criteria, selection biases, definition of outcomes, different quality assessments, and heterogeneous methodology.”).

7) Given your findings, a hot question: what do you think about the possibility of levotiroxine therapy to reduce TSH levels in the prevention of MASLD? I suggest that you add your opinion in the "Discussion" section, possibly including a suggestion for further studies.

Thank you for raising this point. Levothyroxine therapy to reduce TSH levels in the prevention of MASLD has a big issue, deriving from non-selectivity in thyroid hormone receptor (THR) binding, since it binds both THRα and THRβ. The liver mainly expresses THRβ, so levothyroxine could be useful to reduce TSH levels in MASLD prevention, but this would cause hyperthyroidism in other tissues. Indeed, THRα is mainly present in brain, bone, muscle, gut, and heart; TRβ has two different isoforms: TRβ1, which is present in liver and kidney, and TRβ2, expressed in hypothalamus, pituitary, retina, and ear. I added such reflection at page 9, discussion section line 349 (the passage reads: “The drug's selectivity for the target receptor is crucial to mitigate the levothyroxine-induced issue of binding to both THRα and THRβ, which would otherwise lead to hyperthyroidism in non-target tissues.”).

Round 2

Reviewer 1 Report

Comments and Suggestions for Authors

Thanks to the authors for revising and addressing all my recommendations and concerns. The authors have provided point-by-point with in-depth and comprehensive comments on all my recommendations. I have two small comments:

Due to recent revisions of the manuscript, the order of the references has been disrupted. Please fix it.

I suggest considering better ways to present tables 4 more clearly.

Best regards,

Author Response

For research article

Response to Reviewer 1 Comments

1. Summary

Thank you for taking the time to review this manuscript. Please find the detailed responses below and the corresponding revisions/corrections in track changes in the re-submitted files.

2. Questions for General Evaluation

Reviewer’s Evaluation

Response and Revisions

Does the introduction provide sufficient background and include all relevant references?

Yes

Dear reviewer, thank you for your valuable feedback. Please, find our point-by-point response below.

Is the research design appropriate?

Yes

Are the methods adequately described?

Yes

Are the results clearly presented?

Yes

Are the conclusions supported by the results?

Yes

3. Point-by-point response to Comments and Suggestions for Authors

Comments 1:

Thanks to the authors for revising and addressing all my recommendations and concerns. The authors have provided point-by-point with in-depth and comprehensive comments on all my recommendations. I have two small comments:

Due to recent revisions of the manuscript, the order of the references has been disrupted. Please fix it.

I suggest considering better ways to present tables 4 more clearly.

Best regards,

Dear Reviewer 1,

thank you very much for your comments concerning our manuscript entitled " TSH levels are related to fatty liver indices independently of FT4: a cross-sectional study." (Submission ID: jcm-3540666). We fixed the order of the references and presented Table 4 in a clearer way, as per your recommendation.

Thank you again for your comments and suggestions for our manuscript.

Reviewer 2 Report

Comments and Suggestions for Authors

The Authors have revised the manuscript according to the suggestions provided and consequently they have improved it.

Comments on the Quality of English Language

/

Author Response

Dear Reviewer 2,

thank you very much for your comments and suggestions which contributed to improve our manuscript.

Kind regards